# The Interaction between Hypovirulence-Associated Chrysoviruses and Their Host *Fusarium* Species

**DOI:** 10.3390/v16020253

**Published:** 2024-02-05

**Authors:** Chengwu Zou, Xueying Cao, Qiujuan Zhou, Ziting Yao

**Affiliations:** 1State Key Laboratory for Conservation and Utilization of Subtropical Agro-Bioresources, Guangxi Key Laboratory of Sugarcane Biology, College of Agriculture, Guangxi University, Nanning 530004, China; zouchengwu@gxu.edu.cn (C.Z.);; 2Plant Protection Research Institute, Guangxi Academy of Agriculture Science, Nanning 530007, China

**Keywords:** chrysovirus, *Fusarium*, mycovirus–host interactions, viral infection, transmission, RNA interference

## Abstract

Chrysoviruses are isometric virus particles (35–50 nm in diameter) with a genome composed of double-stranded RNAs (dsRNA). These viruses belonged to the *Chrysoviridae* family, named after the first member isolated from *Penicillium chrysogenum*. Phylogenetic classification has divided the chrysoviruses into *Alphachrysovirus* and *Betachrysovirus* genera. Currently, these chrysoviruses have been found to infect many fungi, including *Fusarium* species, and cause changes in the phenotype and decline in the pathogenicity of the host. Thus, it is a microbial resource with great biocontrol potential against *Fusarium* species, causing destructive plant diseases and substantial economic losses. This review provides a comprehensive overview of three chrysovirus isolates (Fusarium graminearum virus 2 (FgV2), Fusarium graminearum virus-ch9 (FgV-ch9), and Fusarium oxysporum f. sp. dianthi mycovirus 1 (FodV1)) reported to decline the pathogenicity of *Fusarium* hosts. It also summarizes the recent studies on host response regulation, host RNA interference, and chrysovirus transmission. The information provided in the review will be a reference for analyzing the interaction of *Fusarium* species with chrysovirus and proposing opportunities for research on the biocontrol of *Fusarium* diseases. Finally, we present reasons for conducting further studies on exploring the interaction between chrysoviruses and *Fusarium* and improving the accumulation and transmission efficiency of these chrysoviruses.

## 1. Introduction

*Fusarium* is a large genus of agronomically important fungi of the *Nectriaceae* family (order *Hypocreales*, class *Sordariomycetes*). *Fusarium* strains are ubiquitous in soil, animals, plants, and organic debris. The pathogenic strains of the *Fusarium* genus are known to infect several plant types, including food crops, cash crops, medicinal plants, and ornamental plants and cause diseases such as root rot, stem rot, stem base rot, flower rot, and ear (grain) rot. These *Fusarium* strains invade the host plant through wounds and enter the vascular system, damaging their transport tissues, producing toxins, and eventually affecting crop yield and quality [1,2]. Although several physical and chemical strategies have been developed, controlling or preventing *Fusarium* diseases is challenging. Several destructive diseases have been reported to be caused by *Fusarium* species globally, such as the wheat *Fusarium* head blight (*F. graminearum*) [3], banana wilt disease (Panama disease; *F. oxysporum*) [4], and sugarcane wilt or top rot (Pokkah boeng disease; *F. sacchari*) [5,6], which result in crop production losses. The authoritative plant mycologists included *F. graminearum* and *F. oxysporum* among the top ten pathogenic fungi in the Molecular Plant Pathology journal in 2012 [7]. These facts suggest the importance of conducting detailed studies managing the plant disease-causing *Fusarium* species [7].

Currently, chemical control is the most commonly used approach for managing fungal diseases. However, using chemicals has led to the development of fungal resistance, caused increased damage to natural ecosystems, and resulted in the accumulation of toxic residues; therefore, safer alternatives, such as the use of biological control agents, have been explored. Mycoviruses, the specialized parasites that proliferate in fungi and result in decreased or no virulence of the host fungi, are considered one of the most promising biocontrol agents [8]. Among them, host hypovirulence-related species of the *Chrysoviridae* family have the potential to be developed as biocontrol agents against *Fusarium*. *Chrysoviridae* is a family of small, isometric, non-enveloped viruses (35–50 nm diameter) with multi-segmented double-stranded RNA (dsRNA). These dsRNA segments are individually encapsidated and constitute a genome of 8.9–16.0 kbp in size; each dsRNA segment is a monocistron. Typically, dsRNA1 encodes an RNA-directed RNA polymerase (RdRp; P1), while dsRNA2 from chrysoviruses with three dsRNA segments or dsRNA3 from chrysoviruses with four or more dsRNA segments encode the main capsid protein (CP; P2 or P3). However, the functions of the proteins encoded by other dsRNA segments remain unknown. Based on molecular phylogenetic analysis using the complete amino acid sequences of RdRp, the chrysovirus isolates cluster into two large groups, classified at the genera level: genus *Alphachrysovirus* includes three or four dsRNA segments and genus *Betachrysovirus* includes four, five, or seven dsRNA segments [9,10]. 

The chrysoviruses are mainly found in Ascomycota and Basidiomycota fungi [9,10,11], and the viruses reported to impact fungal host pathogenicity are mainly of the *Betachrysovirus* genus. These viruses include chrysoviruses mediating the hypovirulence of *Fusarium* strains. Therefore, these mycoviruses have attracted attention as potential biocontrol agents against plant disease-causing *Fusarium* [12,13,14,15]. Due to the hypovirulence effect, Cryphonectria hypovirus 1 (CHV1), found in the plant pathogenic fungus *Cryphonectria parasitica* has been used to control chestnut blight in Europe [16]. However, the effective use of mycoviruses as biological control agents remains challenging. Therefore, in this review, we introduced five chrysoviruses isolated from different *Fusarium* species, focusing on three that inhibit the growth and virulence of the host. We focus on the latest reports on the molecular characteristics, virus–host interaction mechanisms, and the transmission of these three isolates. We also discuss the existing and potential problems and propose plans for research on hypovirulence-associated chrysoviruses to identify effective biocontrol agents against the plant disease-causing *Fusarium* species.

## 2. Chrysovirus Isolates Found in *Fusarium* Species

The first chrysovirus isolated from the Fusarium species was in 2007, Sharzei et al. identified that three dsRNA segments isolated from *Fusarium oxysporum* f. sp. *melonis* of *Cucumis melo* in Iran belonged to the *Alphachrysovirus* genus of the *Chrysoviridae* family [17]. Subsequently, chrysoviruses were isolated from *Fusarium* species infecting crops in various parts of the world. In 2011, Darissa et al. reported a chrysovirus in the ch 9 isolate of the *F. graminearum* strains (China 1–10) obtained from China’s cereals [18]. In the same year, Yu et al. detected the presence of a chrysovirus in *F. graminearum* strain 98-8-60 from the barley plants of Korea and found that the sequences of this virus were highly similar to the one in *F. graminearum* ch 9 isolate from China [19]. In 2015, Lemus-Minor et al. identified four dsRNA segments of chrysovirus from the *F. oxysporum* f. sp. *dianthi* strain Fod166 obtained from the carnation plants of Spain [20]. Later, Yao et al. found chrysovirus in the *F. sacchari* isolate FJ-FZ4 obtained from the sugarcane plants of China [21]. Table 1 summarizes these five chrysovirus isolates reported to inhabit three species of the *Fusarium* genus. 

Among the five Fusarium chrysoviruses, FgV-ch9 and FgV2 showed the highest sequence identity. The putative amino acid sequences of the FgV2 dsRNAs 1 and 3 showed 98% and 99% identity with those of FgV-ch9 [19]. The second-highest sequence identity was detected between FodV1 and FsCV1. The putative amino acid sequences encoded by FsCV1 dsRNA 1 and dsRNA 3 showed 89% and 91% of identity to those from FodV1 [21]. Based on the amino acid sequences of the RdRp and P3 proteins (≤53% amino acid sequence identity in the RdRp and P3), FgV-ch9 and FgV2 were different isolates of a single chrysovirus and FsCV1 and FodV1 were different isolates of another chrysovirus [9]. Interestingly, FgV2 and FgV-ch9 were found in *F. graminearum*, and both resulted in similarly symptoms. On the other hand, FodV1 and FsCV1 were isolated from different *Fusarium* species. Therefore, we speculate that similar to FodV1, FsCV1 may also have a symptomatic effect on its host (*F. sacchari*) strain; however, this aspect needs further examination. Future studies should adopt specific molecular technologies to characterize the functions of the multi-segments of Fusarium chrysovirus isolates to realize similarities and dissimilarities in their effect on the host. 

Furthermore, a comparison of the dsRNA segments of the chrysoviruses revealed that their 5′ UTRs were ~62% identical, contained two stretches of identical sequences of 10–14 nt and 23–42 nt long, and were rich in CAA [19,20]. Similar (CAA)n repeats have been identified as the enhancer elements at the 5′ UTRs of tobamoviruses [22]. On the other hand, the 3′ UTRs of the genomic dsRNAs of the above chrysoviruses contained less conserved sequences and were relatively conserved within the dsRNAs of the same species but not among different species. Researchers also found duplications in the 3′ UTR of dsRNA 2 and 3 of FgV-ch9 and FgV2, arranged in a head-to-tail manner and attached to the complete terminus [23]; however, these duplications did not affect the encoded gene. These reports implied that dsRNA 2 and 3 are located close to one another but get transported into the cytoplasm after transcription [23].

## 3. Effects of Chrysovirus on *Fusarium* Species

Studies have reported that chrysoviruses, especially the members of the *Betachrysovirus* genus, such as FgV-ch9, FgV2, and FodV1, influence the colony phenotype, growth, conidiation, sporulation, and pathogenicity of the *Fusarium* host. These impacts have been reported to be proportional to the viral load. At very low levels of FgV-ch9 dsRNAs (detected via RT-PCR but not visible on electrophoresis gel), the phenotype of the *F. graminearum* ch 9 strain remained unchanged. But, as the levels of FgV-ch9 dsRNAs in the *F. graminearum* host increased (visible dsRNA bands), the mycelial growth rate and conidiospores yield decreased, the colonies morphology became more abnormal, the cytoplasm became disordered, and the virulence decreased [24]. The effect of FgV2 on *F. graminearum* strain 98-8-60 was similar to that of FgV-ch9 on its host. Moreover, FgV2 infection of *F. graminearum* led to defects in perithecium development, impacting sexual reproduction [25]. FodV1 also significantly reduced the vegetative growth, conidiation rate, and pathogenicity of the fungal host [26]. Besides, FodV1 in the *F. oxysporum* strain 116 of carnation plants reduced the host mycelia’s colonization speed and spatial distribution in the plant [27]. Studies also reported differences in the progression of plant disease caused by the virus-infected and uninfected *Fusarium* strains. Though the infected and the uninfected *F. oxysporum* strains colonized the external roots of the carnation plants similarly in the initial stages, the virus-free strain colonized the internal tissues faster than the FodV1-infected strain. Subsequently, the hyphae of the virus-free and FodV1-infected strains progressed through the xylem vessels and reached the central medulla. At this stage, the virus-free strain densely colonized the xylem vessels and the interior of the medulla cells, while the virus-infected strain colonized the xylem vessels only at a lower density and appeared mainly in the intercellular spaces of the medulla cells [27]. 

Importantly, the impact of FgV-ch9 on the host at the subcellular level has been determined through microscopic observation; the effects on subcellular ultrastructure were also proportional to the viral load [24]. Most cells of the strain ch9 with low dsRNAs level appeared normal and had organized cytoplasm and nuclei with smooth nuclear membranes, similar to those without dsRNA. Meanwhile, the fungal cells with medium dsRNAs levels had normal nuclei and abundant ribosomes, with partial cytoplasmic disorganization. However, when FgV-ch9 dsRNAs were high in the fungal cells, abnormal cytoplasm appeared with many large vacuole-like structures, as well as disintegrated nuclear and mitochondrial membranes [24]. These changes in the subcellular ultrastructure and macroscopic phenotype of the fungal host indicated that the stage when the viral load attained a medium level in the host cell might be crucial. It seemed that FgV-ch9 might successfully overcome the host’s defense mechanism at this stage, and the dsRNAs could attain a medium load, resulting in partial cytoplasmic disorganization with normal nucleus and abundant ribosomes. At this stage, a visible negative impact was also detected on the mycelial growth and virulence of the fungal host. Thus, the findings suggest that future studies aiming to identify an effective biocontrol agent should focus on analyzing this crucial stage.

## 4. Interaction of *Fusarium* Species with Chrysoviruses

The host activates defense mechanisms against viral infection through transcription factors (TFs). The virus interferes with and controls host TFs against the host defense system and makes use of host TFs for replication. In this regard, typically, transcription factors (TFs) directly or indirectly regulate defense mechanisms by activating or repressing a series of genes or pathways in the interaction of virus and host [28]. In *F. graminearum*, a series of TFs have been shown to be related with the viral accumulation of Fusarium graminearum virus 1 and changes in host phenotype and response to stress [29]. Therefore, identifying TFs associated with viral infection is of great significance for understanding the interaction between the chrysovirus and the *Fusarium* host. An investigation of the differences in the global gene expression patterns using RNA-seq revealed that 37 TFs, including 31 upregulated and 6 downregulated ones, were differentially expressed in *F. graminearum* infected by FgV2 compared with the uninfected strain. Among the upregulated TFs genes, 21 belonged to the Zn2Cys6 family, indicating that FgV2 might utilize the host Zn2Cys6 TFs which might be associated with defense response against FgV2 for their replication [30]. 

Further, Kwon et al. introduced FgV2 into a TF gene deletion mutant library (657 TF gene deletion mutants) of *F. graminearum* strain GZ03639 to identify the TFs associated with the FgV2-host interaction [31]. The specific TFs were identified as antiviral factors against FgV2 replication since the mutants had higher FgV2 accumulation than the infected wild-type strain. Meanwhile, as some mutants infected by FgV2 restored the phenotype as the infected wild-type strain, the TFs were identified as proviral factors. Detailed analysis revealed that these TFs identified as proviral or antiviral factors mainly belonged to bZIP, GATA-type zinc finger, and Myb families. A few TFs were also involved in FgV2 accumulation and defective interfering RNA (DI-RNA) generation. When DI-RNAs and reduced viral dsRNA load were detected, the mycelial growth rate increased in these TF mutants. DI-RNA were small segments generated from FgV2 RNA3; however, whether the DI-RNA function as a symptom development factor and eventually inhibit virus replication remains uncertain. A few other TFs were identified to be involved in siRNA interference, enhancing DNA damage- or reactive oxygen species (ROS)-responsive pathways and components of the host in response to FgV2 infection. All these results provided broader insights into the TF-mediated FgV2- *F. graminearum* interaction [31]. 

A few additional fungal factors have been identified as regulating the fungal host–virus interaction. Bormann et al. found that a putative fungal mRNA-binding protein was related to FgV-ch9-related symptom severity in *F. graminearum* [32]. The expression of the gene encoding this putative fungal mRNA binding protein named viral response 1 (vr1, locus tag FGSG_05737) was downregulated in response to FgV-ch9 infection. Deleting *vr1* in *F. graminearum* strain PH1 resulted in symptoms similar to those infected by FgV-ch9, while constitutive expression of *vr1* in the FgV-ch9 infected PH1 exhibited an infection-free phenotype. These observations suggested that the putative fungal mRNA binding protein negatively regulates chrysovirus infection. Interestingly, the constitutive expression of FgV-ch9 structural protein P3 triggered the downregulation of *vr1* and resulted in symptoms like FgV-ch9 infection [32]. However, detailed studies must reveal the interactions between vr1 and P3 and their role in the fungal regulation of viral infection. The findings of those studies will help identify candidates to manipulate the chrysoviral infection in *Fusarium* and manage plant diseases. 

Research has also suggested the role of various processing mechanisms in regulating fungus-chrysovirus interaction. In 2011, an analysis of FgV-ch9 RNA2 and RNA3 sequences indicated that the proteins P2 and P3 were 94 kDa and 93 kDa, respectively [24]. However, electrophoresis showed that P2 was 62 kDa and P3 was 70 kDa. Later, Lutz et al. purified the viral proteins as polysomes and demonstrated that P2 and P3 of FgV-ch9 were expressed as full-length proteins in infected mycelia of three-day old cultures. However, during maturation, the P2 and P3 proteins underwent proteolytic processing. The C-terminal domain of P3 was degraded entirely after capsid processing and particle assembly [33]. Further elucidation of the steps involved in protein processing will provide novel insights into the interaction between the virus and the fungal host.

## 5. RNA Interference in *F. graminearum* Responding to FgV2 Infection

In eukaryotes, RNA interference (RNAi) is a major defense response to viruses. Information on the RNAi pathway of fungi is crucial for assessing the interaction with the virus. The RNAi pathway uses small, non-coding RNAs to regulate endogenous genes and defend against viral infection. The classical RNAi pathway uses a set of host components, including Dicers, Argonautes (AGO), and RdRps. Dicers recognize viral dsRNA and cleave it to produce small RNA (sRNA), also called small interfering RNA (siRNA) or microRNA (miRNA), or virus-derived small RNAs (vsiRNAs) that are 21 to 24 nucleotides (nt). One strand of the double strand sRNA binds to the AGO protein to form the functional RNA-induced silencing complex (RISC), which can bind to complementary sequences and specifically cleave target RNA. Lee et al. found that when *F. graminearum* strain PH-1 was infected by FgV2, the genes encoding RNAi components, *FgDICER-1* and *-2*, *FgAGO-1* and *-2*, and *FgRdRp-3*, *-4*, and *-5* expression was upregulated, demonstrating the host response to FgV2 infection [25]. However, when these genes of RNAi components were single deleted and then infected with FgV2, the viral load in the single gene deletion mutants were similar to that in the FgV2-infected wild-type strain, and the phenotype of these mutants was still similar to the phenotype of FgV2-infected wild-type strain. These observations indicated that disrupting a single gene among the RNAi components may not directly interfere with the antiviral response of *F. graminearum* [34]. However, when FgV2 was introduced into *FgDICER*s or *FgAGO*s double gene knockout mutants, their growth appeared retarded more than that of the single gene mutant. Compared with the FgV2-infected wild-type strain, the double gene knockout mutant exhibited increased accumulation of FgV2. These results indicated that *FgDICER* and *FgAGO* play redundant roles during the response of *F. graminearum* to FgV2 infection [34]. A high-throughput sequencing of the siRNA library obtained from the FgV2 infected PH-1 strain showed that the vsiRNA reads were 18 to 24 nt long; here, vsiRNAs with 20 to 22 nt in length dominated, and the percentage of vsiRNA reads in the sense strand was significantly higher than that in the antisense strand. Moreover, the percentage of vsiRNA reads varied among the five segments of FgV2, with a relatively high percentage in the dsRNA3 segment. In dsRNA1, the vsiRNAs reads were higher in the 5′ and 3′ UTRs of the sense strand and the internal region of the antisense strand. In dsRNA2, -4, and -5, vsiRNAs were dispersed throughout the segments. Studies also showed that the dsRNA3 segment had more vsiRNA hotspots than the other four segments. These results indicated that the dsRNA3 may be preferentially recognized and targeted by the fungal RNAi components during defense response [34]. It also suggested that further exploration of RNA3 while analyzing the factors influencing FgV2 accumulation is required to reveal the fungal defense response to the chrysovirus.

## 6. Transmission of Fusarium Chrysoviruses

Generally, most fungal viruses are transmitted vertically via host asexual or sexual spore production or horizontally via hyphal anastomosis [8]. The effectiveness and success of biocontrol agent application are correlated to the transmission rate of mycoviruses. The higher the horizontal transmission rate, the more extensive the impact range of the virus [35]. Similarly, the higher the vertical transmission rate, the longer the effectiveness of the virus’s impact [35]. Thus, understanding and improving the transmission rate of mycoviruses will enhance their effectiveness and success as biocontrol agents. Scientists could successfully introduce FgV2 and FgV-ch9 from the host strain to the same or derived mutant free of virus through hyphal anastomosis or protoplast anastomosis [31,34], but there has been little research on whether they may be transmitted between strains in different vegetative incompatible groups, species, or even genera. Lemus-Minor et al. found that FodV1 was horizontally transferred between compatible isolates in the lab through hyphal anastomosis, attaining high accumulation in the recipient isolate; this virus was also transmitted vertically during sporogenesis [36]. However, a detailed analysis of the vertical transmission rate showed that one FodV1-infected isolate produced 24% conidiospores carrying dsRNA, while another produced 100% conidiospores carrying dsRNA [36]. Since the vertical transmission efficiency varies among the FodV1 isolates, future work should compare these isolates, explore the factors affecting their vertical transmission efficiency, and identify effective agents. It has also been found that the vertical transmission efficiency of FgV2 is affected by the generation of DI-RNA. In the host strain, the vertical transmission efficiency of FgV2 without DI-RNA from the first generation to the third generation was as high as 100%. However, FgV2 in the host strain with DI-RNA generation demonstrated a low vertical transmission efficiency (14.3–25%) from the first generation to the third generation, with constant production of DI-RNA in all three generations [31]. Contrary to FgV2, the vertical transmission of Trichoderma harzianum hypovirus 1 (ThHV1) was prompted by its DI-RNA (ThHV1-S) when ThHV1 and ThHV1-S existed together in conidiospores of *Trichoderma harzianum* [37]. Furthermore, several TF deletion mutants of the *F. graminearum* GZ03639 strain have demonstrated FgV2 transmission via protoplast fusion but not via hyphal anastomosis, indicating the role of TFs in hyphal anastomosis-related processes, such as regulating vegetative incompatibility [31]. Therefore, future studies should address the abovementioned diverse aspects to improve the horizontal and vertical transmission rates of the Fusarium chrysoviruses for developing effective biocontrol agents.

## 7. Conclusions and Prospects

Diseases caused by *Fusarium* species are a huge threat to global crop production. Though research on chrysoviruses infecting *Fusarium* species has progressed, some challenges must be overcome. Specifically, the transmission of chrysoviruses needs to be addressed to enhance their efficacy as biological control agents [37]. Moreover, the interaction between chrysovirus and *Fusarium* remains unclear. A lack of understanding about these aspects has delayed the application of chrysoviruses in preventing and controlling *Fusarium* diseases. 

Furthermore, scientists should aim to clarify the key factors that impact chrysoviral accumulation in *Fusarium* species (Figure 1). Studies have suggested that an increased load of hypovirulence-associated chrysoviruses significantly affects the *Fusarium* host, and therefore, it is crucial to identify the factors that impact chrysoviral accumulation. Multi-omics technologies combining transcriptomics, proteomics, and metabolomics should be used to compare the *F. graminearum* strains with different viral accumulations and explore key genes or pathways and molecular mechanisms, providing theoretical and technical support. Researchers have also shown that the DI-RNAs interfere with FgV2 accumulation, restoring the phenotype and pathogenicity of *F. graminearum*. In addition, expression of the FgV-ch9 structural protein P3 caused virus infection-like symptoms when expressed in the wild type strain of *F. graminearum* [32]. Therefore, mutants of these proteins will serve as important resources for exploring key factors in chrysoviral accumulation. In addition, research on Fusarium chrysovirus should not be limited to the ORF region of dsRNAs. Studies have suggested that 5′ UTR with high conservation and rich in CAA might contain enhancer sequences, while 3′ UTR with less conservation might be associated with viral protein processing and stability. However, there are few reports on 5′ UTR and 3′ UTR, and further studies should clarify their role in virus–host interaction.

Identifying the key factors regulating the spread of Fusarium chrysoviruses is another major aspect that demands research (Figure 1). Currently, only five isolates of three *Chrysoviridae* species are found in *Fusarium* species, and their distribution is not widespread. Among these, only two isolates, FgV-ch9 and FsCV1, were discovered in China, each from a different *Fusarium* species and found only in a single location. These observations suggest a low transmission rate of these two isolates, which results in sparse distribution. However, this assumption should be verified. Moreover, research should focus on improving the hypovirulence-associated chrysoviruses (FgV-ch9, FgV2 and FodV1). Analysis of the published works also suggested that advanced technologies using chemicals should be adopted to break through vegetatively incompatible barriers and improve the spread of the Fusarium chrysovirus isolates. In 2013, Ikeda et al. found that zinc compounds weakened the vegetative incompatibility and improved the transmission efficiency of Rosellinia necatrix megabirnavirus 1 between *Rosellinia necatrix* strains [38]. The mycoviruses have also been found to be transmitted among strains via genetic transformation technologies in the lab. Specifically, protoplast fusion, electroporation, and polyethylene glycol (PEG)-mediated methods were used to transfer viral particles/RNA/viral cDNA into fungal protoplasts and obtain replicable viruses, improving their spread between strains or even species [39,40,41]. Zhang et al. used genetic techniques to knock down the vic (vegetative incompatibility) loci of *C. parasitica* strains and break down the vegetative incompatibility; this approach engineered the strains to serve as super donors of hypovirus in the field [42,43]. In addition, Liu et al. found that Sclerotinia sclerotiorum hypovirulence-associated DNA virus 1 (SsHADV-1) could be transferred to other *S. sclerotiorum* strains through the mycophagous insect *Lycoriella ingenua* [44]. Thus, exploring the transmission vectors is another way to improve viral spread.

More importantly, research on efficiently utilizing fungal viruses to prevent and control plant diseases should be strengthened (Figure 1). We should apply advanced biological technologies and focus on exploring Fusarium viral resources. The primary source of mycoviruses is the *Fusarium* species obtained from diseased plant tissues. Generally, these fungal strains are highly pathogenic and possess a low possibility of carrying hypovirulence-associated mycoviruses. Therefore, we should concentrate on isolating and cultivating non-virulent or less virulent *Fusarium* strains from diverse ecological niches, such as soil and plots with low disease incidence, which is difficult and time-consuming. In addition, independently of the isolation and purification techniques, metatranscriptomics should be used to identify the viral sequences in the environmental samples to further accelerate the mining of mycovirus resources. Meanwhile, novel ideas can be planned for current research after identifying new viruses. Moriyama et al. found that the viral particles of MoCV1 were released into the culture supernatant and directly infected the closely related fungal colonies [45]. In addition, successful expression of its P3 gene affected various physiological mechanisms in yeast cells, leading to symptoms similar to those caused by viruses [45,46]. These results suggest that if a single gene of mycovirus could play a significant role in the host fungus, it may have great potential in generating effective biological control agents. Thus, understanding mycoviruses, improving genetic technologies, and utilizing significant genes might broaden their application. 

## Figures and Tables

**Figure 1 viruses-16-00253-f001:**
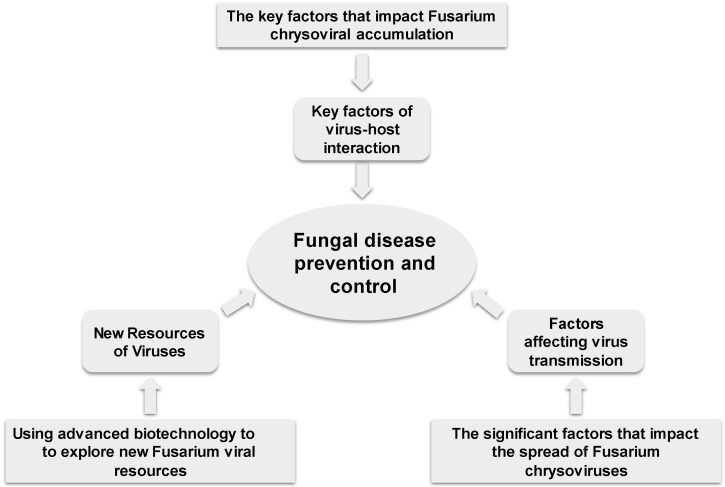
Strategy diagram for fungal disease prevention and control using Fusarium chrysoviruses.

**Table 1 viruses-16-00253-t001:** Fusarium chrysoviruses.

Chrysovirus Name	Genus	Host	Effects on Host	dsRNA Segment (nt)	Size (nt)	5′ UTR (nt)	3′ UTR (nt)	Accession No.	Coding Protein	Plant	Country	References
Fusarium oxysporum chrysovirus 1 (FoV1)	*Alphachrysovirus*	*F. oxysporum* f. sp. *melonis*.	unknown	RNA1	2574 *	—	—	EF152346	RdRp	Melon	Iran	[17]
				RNA2	648 *	—	—	EF152347	P2			
				RNA3	994 *	—	—	EF152348	nonfunctional putative protease			
Fusarium graminearum mycovirus-China 9 (FgV-ch9)	*Betachrysovirus*	*F. graminearum* strain China 9	Mycelia growth ↓ Pigmentation ↑ Conidiation ↓ Pathogenicity ↓	RNA1	3581	82	84	HQ228213	RdRp	Cereals	China	[18]
				RNA2	2931	93	210	HQ228214	P2			
				RNA3	3002	105	326	HQ228215	P3			
				RNA4	2746	78	160	HQ228216	P4			
				RNA5	2928	96	270	HQ228217	Contains a C2H2 zinc finger domain			
Fusarium graminearum virus 2 (FgV2)	*Betachrysovirus*	*F. graminearum* strain 98-8-60	Mycelia growth ↓ Pigmentation ↑ Conidiation ↓ Pathogenicity ↓	RNA1	3580	82	84	HQ343295	RdRp	Barley	Korea	[19]
				RNA2	3000	93	279	HQ343296	P2			
				RNA3	2982	105	306	HQ343297	P3			
				RNA4	2748	78	162	HQ343298	P4			
				RNA5	2414	97	184	HQ343299	Contains a C2H2 zinc finger domain			
Fusarium oxysporum f. sp. dianthi mycovirus 1 (FodV1)	*Betachrysovirus*	*F. oxysporum* f. sp. *dianthi* strain 116	Mycelia growth ↓ Conidiation ↓ Pathogenicity ↓	RNA1	3555	82	53	KP876629	RdRp	Carnation	Spain	[20]
				RNA2	2809	84	88	KP876630	P2			
				RNA3	2794	97	138	KP876631	P3			
				RNA4	2646	97	56	KP876632	P4			
Fusarium sacchari chrysovirus 1 (FsCV1)	*Betachrysovirus*	F. sacchari strain FJ-FZ04	unknown	RNA1	3518	63	35	MN295964	RdRp	Sugarcane	China	[21]
				RNA2	2796	72	87	MN295965	P2			
				RNA3	2779	83	137	MN295966	P3			
				RNA4	2569	39	31	MN295967	P4			

* Incomplete sequence.

## Data Availability

No new data were generated in this work.

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
