# Peer review of "The Interaction between Hypovirulence-Associated Chrysoviruses and Their Host Fusarium Species"

_viruses, 2024, doi:10.3390/v16020253_

Round 1

Reviewer 1 Report

Comments and Suggestions for Authors

You appear to have conducted a thorough review of the literature of Chrysoviruses in Fusarium sp., particularly focusing on the FgV2, FGV-ch9, and FodV1. The scope of the review includes virus-host interactions, host immune response, virus transmission, and commentary about future directions of research related to agronomy. The topic is interesting and important.

Overall, I think the content of the paper is worthwhile and will be a welcome addition to the literature on mycoviruses after some attention is paid to the writing.

Comments on the Quality of English Language

My most substantial comments are about the writing, though I also note some areas of scientific differences in interpretation in the attachment. Overall, the reader would greatly benefit from some restructuring of the content. There are also some major issues with readability in English. The authors should not be faulted on this, as it is clear that English is not their first language! Therefore, I have taken great care in documenting in line-detail alterations to the writing that will improve the English, making it more readable. I completed this for much of the paper, but was not able to do for the entirety. I do hope this is useful and that the authors can themselves improve the remainder of the article. I hope the authors can find resources for assistance with writing for future works prior to submission.

Reviewer 2 Report

Comments and Suggestions for Authors

In this work, authors summarized the recent advances in studies of chrysoviruses on regulation of Fusarium response to virus infection, RNA interference, and transmission, providing a reference for analyzing the pathogenic mechanism of Fusarium spp. and discussing opportunities for research on the biocontrol of Fusarium diseases. Unfortunately, the Manuscript has no advantages comparing to the published similar review of Li P, Bhattacharjee P, Wang S, Zhang L, Ahmed I and Guo L. Mycoviruses in Fusarium Species:An Update. Front. Cell. Infect. Microbiol. 2019 9:257.doi: 10.3389/fcimb.2019.00257. No appealing ideas were showed in this manuscript. Therefore, I think this review might not be published in the Viruses journal.

Reviewer 3 Report

Comments and Suggestions for Authors

The following review is dedicated to the currently identified chrysoviruses that infect the Fusarium fungi. The issue of agricultural crops being affected by these fungi is big and highly relevant, making the search for potential methods to combat them of great interest to many researchers.
Nevertheless, I have several comments and additions to the presented review:
1.  I consider it necessary to include the term “viral infection” to the keywords list.
2.  In the section "2. Five chrysovirus isolates found in Fusarium spp.," below the table is a description of five viral isolates. However, this information completely duplicates what is already presented in the table, so I believe this text is redundant and should be removed. The table alone is preferable as it’s comprehensive.
3.  In the "7. Conclusion and prospects" section, it would be nice to include a general scheme of anticipated actions—a figure that concisely reflects the course of further researches.

Reviewer 4 Report

Comments and Suggestions for Authors

Authors Chengwu Zou and co-workers presented here a manuscript entitled "Chrysoviruses in Fusarium species".

The authors prepared a comprehensive review of the current knowledge in the field of mycoviruses, especially chrysoviruses, focusing on five important chrysoviruses infecting fungi of the genus Fusarium. The selected chrysoviruses have the potential to reduce the pathogenicity of plant pathogenic fungi.

The topic is original and innovative, with a strong practical aspect.

The present review summarises the current knowledge in this field. I am not aware of any other review on a similar topic.

A review manuscript with the stated focus covers the intended topic well.

The review contains 21 out of 51 citations that are not older than five years.

There are an acceptable number of self citations.

The statements and conclusions are coherent and supported by the cited references.

The only table summarises the known chrysoviruses infecting Fusarium fungi.

In a table, genera should be written as one word (e.g. alphachrysovirus, betachrysovirus).

The manuscript needs more detailed introduction of viruses including reorganising some chapters. For example, lines 373-391 – the entire paragraph about sequence differencies belongs to the beginning to introduce viruses of interest.

Minor comments:

lines 59-60: The virus isolates belonging to the Chrysoviridae family are small, isometric, non-enveloped particles: use either „virus isolates HAVE small....particles“  or „the viruses belonging to the....“ (without „isolates“)

line 65: main capsid protein (CP; P2 or P3) - explain the „P2 or P3“

lines 73-80: sentences need better structure and connection as they are logical continuations

line 99: chrysovirus reported to BE isolated from

lines 114-117: you cannot write that the genome contains incomplete RNAs. They are simply not fully sequenced.

line 137: genu - genus

line 187: lever - level?

lines 239-242 belongs more to Conclusion

lines 290-291: „few research on they also could be transmitted“ does not make sense

line 316: chrysoviurses - chrysoviruses

line 328: „...strain cannot either answer this question. The „either“ belongs at the end of the sentence

line 358: above fore ?

lines 434-435 Funding: "please add" doesn't seem to belong here.

lines 436-445 Institutional Review Board Statement: the whole paragraph doesn't belong here.

Comments on the Quality of English Language

The English used in the manuscript is mostly grammatically correct. However, the meaning of the text is sometimes unclear.

Round 2

Reviewer 1 Report

Comments and Suggestions for Authors

The authors have adequately addressed the comments. 

Reviewer 3 Report

Comments and Suggestions for Authors

The revised version of the manuscript contains all the necessary corrections that I've advised in the first round of the review.